# Comparative analysis of gut microbiota in healthy and diarrheic foals

Di Zhu,[1] Siyu Li,[1] Zhixiang Xu,[1] Md. F. Kulyar,[1] Xu Bai,[2] Yu Wang,[2] Boya Wang,[1] Emaan Khateeb,[1] Dandan Deng,[1] Lidan Wang,[1] Yuji Chen,[1] Aizhen Guo,[1] Yaoqin Shen[1]

**ABSTRACT** Diarrhea presents a substantial risk of high morbidity and mortality among foals. Although studies have shown connections between gut microbiota and several gastrointestinal diseases, there is still inadequate information on gut microbial alterations in foals during diarrhea. In this study, we conducted 16S rRNA and ITS gene amplicon sequencing to investigate gut bacterial and fungal differences between healthy and diarrheic foals. The results unveiled significant reductions in gut bacterial and fungal diversities among foals experiencing diarrhea, accompanied by notable shifts in the composition of gut microbial communities. A considerable decrease was observed in the relative abundance of 30 bacterial and 34 fungal genera. Moreover, two bacterial and eight fungal genera were utterly undetectable in the gut microbiota of diarrheic foals. Some decreased genera, such as *Bifidobacterium* and *Saccharomyces*, were deemed beneficial and recognized as probiotics. The study revealed significant alterations in foals' gut bacterial and fungal communities during diarrhea, which enriched our comprehension of gut microbial dynamics in foals across varying health statuses. These findings offer valuable insights for managing diarrhea through gut microbiota modulation, suggesting that probiotics may be superior to antibiotics in preventing and controlling foal diarrhea.

**IMPORTANCE** This research advances the understanding of gut bacterial and fungal dynamics in foals, highlighting gut microbiota dysbiosis as a potential contributor to foal diarrhea. Additionally, we observed that many altered bacteria and fungi were downregulated during diarrhea, including some probiotic strains. Consequently, our findings provide evidence that probiotics may offer superior efficacy compared with antibiotics as potential candidates for preventing and treating foal diarrhea.

**KEYWORDS** foal, diarrhea, gut microbiota, 16S rRNA, ITS

Diarrhea significantly impacts animal productivity, immunity, and even mortality, resulting in substantial economic losses over hundreds of billions of dollars per year (1, 2). Early studies suggested that diarrhea affects almost all species, with neonatal animals being more susceptible, posing significant challenges to livestock husbandry (3–5). In horses, foal diarrhea affects up to 60% of young horses during the first 6 months of life, often without a clearly identifiable cause (6, 7). Modern research on diarrhea has shifted the emphasis from harmful bacteria, parasites, and viruses to the particular gut microorganisms that go through periods of dominance and decline (8–10). Preliminary studies have shown that fecal microbiota transplantation (FMT) may reduce diarrhea symptoms, emphasizing the crucial role of gut microbiota in preventing this condition (11–13).

The gut microbiota consists of trillions of microorganisms, outnumbering host cells by approximately 10-fold (14). More than 98% of all gastrointestinal microorganisms

**Peer Reviewer** Yuchao Zhao, Beijing University of Agriculture, Beijing, China

Address correspondence to Yaoqin Shen, yshen@mail.hzau.edu.cn.

The authors declare no conflict of interest.

See the funding table on p. 14.

are bacteria, with fungi accounting for 0.1%–1%, whereas viruses and protozoa round out the rest (15, 16). Intricate network communication between the host and these microorganisms is essential for metabolic processes, nutrition absorption, immunological regulation, and gut barrier maintenance (17–20). Beneficial gastrointestinal bacteria and fungi can inhibit the colonization of foreign pathogens by regulating the local ecological environment, secreting antimicrobial peptides, and competing for nutrients under normal physiological conditions (21). However, gut-residing opportunistic pathogens, which are usually considered components of the gut microbiota, can trigger illness by exploiting situations of gut microbial dysbiosis (22). Considering a symbiotic system, the gut microbiota is regarded as a vital organ influencing host health (23). Numerous studies have demonstrated that gut microbiota dysbiosis can lead to gastrointestinal disorders, such as diarrhea, constipation, and enteritis (24, 25). Furthermore, increasing evidence suggests that gut microbial alterations can extend the adverse effects beyond the gastrointestinal system, impairing the functions of other organs, including the liver and brain (26, 27).

High-throughput sequencing technology has proven effective in investigating changes in the gut microbial composition following the onset of various diseases. This technological advancement facilitates a comprehensive examination of the potential relationship between gut microbiota and specific illnesses (21, 28). Furthermore, conducting in-depth analyses of the complex gut microbiota can enhance our understanding of the mechanism of certain diseases, enabling strategies to minimize associated collateral damage (29). Research on the role of gut microbiota in diarrhea has been well established in several species, including piglets, giraffes, yaks, and goats (5, 9, 10, 30). Still, there is a lack of research on foals, particularly regarding the gut fungal community. A previous culture-based study found no distinction between diarrheic and healthy foals in the gut bacterial community. However, culture-based techniques have limited capacity to thoroughly understand the complexities of the gut microbiota (31). Given this gap, we investigated healthy and diarrheic foals' gut bacterial and fungal compositions using high-throughput sequencing technology to deepen our understanding of the underlying connection between gut microbiota and foal diarrhea.

## MATERIALS AND METHODS

### Animals and sample collection

In this study, a total of 10 6-month-old free-range Mongolian horses, comprising five healthy and five diarrheic, were selected for sample collection. All foals were raised under identical conditions at the Yumayuan Scenic Area, Inner Mongolia Autonomous Region, China. A trained veterinarian diagnosed diarrheic foals with non-infectious diarrhea and did not provide any treatment until the end of the sample-collecting period. On the day of sample acquisition, all foals were placed in separate pens to prevent potential cross-infection and sample contamination. Fresh fecal samples were collected immediately after defecation. To further minimize contamination, the feces were then resampled from the intermediate portion (approximately 100 g). The freshly obtained samples were placed into sterile plastic containers and promptly stored at −80°C for further analysis.

### DNA extraction, amplification, and sequencing

Both bacterial and fungal genome DNA were extracted from the fecal samples using the TGuide S96 Magnetic Stool DNA Kit (TiangenBiotech, China) following the manufacturer's instructions. Stool samples of 500 µg each were combined with 250 µg of grinding beads and subjected to grinding for 15 min. The resulting mixture was then centrifuged at 12,000 rpm for 1 min, and genomic DNA was extracted from the supernatant using the magnetic bead method. The concentration of extracted genomic DNA was measured with the Qubit dsDNA HS Assay Kit and Qubit 4.0 Fluorometer (Thermo Fisher Scientific, United States), whereas the quality was assessed using a UV-Vis spectrophotometer

(NanoDrop 2000, United States), and the integrity was evaluated by 0.8% agarose gel electrophoresis. Specific primers were synthesized for the bacterial 16S rRNA gene (338F: 5′-ACTCCTACGGGAGGCAGCA-3′ and 806R: 5′-GGACTACHVGGGTWTCTAAT-3′) and the fungal ITS gene (ITS2F: 5′-GCATCGATGAAGAACGCAGC-3′ and ITS2R: 5′-TCCTCCGCT TATTGATATGC-3′) to amplify the V3/V4 and ITS2 conserved regions from the genomic DNA. The PCR reaction mixture consisted of 4 µL of Fast Pfu buffer, 2 µL of dNTPs (2.5 mM), 0.8 µL of each primer (5 µM), 0.4 µL of Fast Pfu polymerase, 10 ng of template DNA, and ddH$_2$O to a final volume of 20 µL. The PCR amplification conditions were as follows: an initial denaturation at 95℃ for 3 min, followed by 27 cycles of denaturation at 95℃ for 30 s, annealing at 55℃ for 30 s, and extension at 72℃ for 45 s, with a final extension at 72℃ for 10 min, concluding at 4℃. The PCR amplicons were purified using Agencourt AMPureXP Beads (Beckman Coulter, United States) and quantified using the Qubit dsDNAHS AssayKit and Qubit 4.0 Fluorometer (Thermo Fisher Scientific, United States). The final products were used to construct the sequencing library using the TruSeq Nano DNA LT Library Prep Kit (Illumina, United States). Purifying libraries, fluorescent quantitation, quality control, and sequence repair followed manufacturer's protocol recommendations. Only the libraries that passed the quality inspections were diluted and denatured for high-throughput sequencing using Illumina novaseq 6000 (Illumina, United States).

## Bioinformatics and statistical analysis

The raw sequencing data were initially filtered by Trimmomatic (v0.33) based on the single nucleotide quality, and primer sequences were identified and removed by Cutadapt (v1.91). Subsequently, the processed data were assembled using USEARCH (v10), and the chimeras were eliminated with UCHIME (v8.1). Similar sequences were clustered into the same operational taxonomic unit (OTU) by USEARCH (v10.0), and the OTUs were annotated at various taxonomic levels using the SILVA database (release 132). Alpha diversity indices, including ACE, Chao1, Shannon, and Simpson, were calculated by QIIME2. Beta diversity analyses, including PCoA and UPGMA, were performed by QIIME. Additionally, linear discriminant analysis (LDA) and its effect size analysis (LEfSe) were employed to identify significantly abundant taxa (from phylum level to genus level) among distinct groups.

Statistical analysis of alpha diversity and differential microorganisms was performed using GraphPad Prism (v8.0), whereas the remaining data were analyzed using the R package (v3.0.3). Sequences obtained were clustered into OTUs with a sequence similarity threshold of over 97%. Differences between groups in the alpha diversity analysis were evaluated using Student's $t$-test, whereas the Wilcoxon rank-sum test was applied to assess significant changes in the relative abundance of gut microorganisms. Additionally, bacterial taxa were identified as significantly abundant between groups based on LEfSe analysis, with an LDA score greater than 4.0. The results are presented as means ± standard deviations, with statistical significance set at $P < 0.05$.

## RESULTS

### Sequence analysis

Following amplicon sequencing, a total of 785,453 (CF = 404,502, DF = 380,951) and 740,222 (CF = 335,467, DF = 404,755) raw sequences were obtained from the bacterial V3/V4 and fungal ITS2 conserved regions, respectively (Table 1). The raw sequences underwent evaluation and filtering, resulting in 1,192,235 effective reads, with an average amount of 546,973 (ranging from 44,579 to 61,503) and 645,262 (ranging from 45,434 to 74,487) high-quality sequences from the bacterial and fungal populations per sample, respectively (Table 2). Both rarefaction and Shannon-index curves of each sample demonstrated a tendency to saturate, indicating sufficient sequencing depth for further analysis (Fig. 1A, B, G and H). The rank-abundance curves of all samples exhibited a wide length and slow decline, suggesting satisfactory evenness and abundance (Fig.

**TABLE 1** The bacterial sequence data of each sample

| Sample | Raw reads | Clean reads | Denoised reads | Merged reads | Effective reads |
|--------|-----------|-------------|----------------|--------------|-----------------|
| CF1 | 87,357 | 78,486 | 75,569 | 67,658 | 61,503 |
| CF2 | 82,342 | 73,716 | 70,670 | 62,940 | 56,146 |
| CF3 | 79,623 | 72,104 | 69,691 | 62,026 | 55,844 |
| CF4 | 87,558 | 78,608 | 76,348 | 68,611 | 61,193 |
| CF5 | 67,622 | 60,662 | 59,642 | 53,148 | 48,313 |
| DF1 | 80,312 | 72,793 | 69,346 | 61,902 | 55,453 |
| DF2 | 65,252 | 58,848 | 55,987 | 49,820 | 44,579 |
| DF3 | 80,100 | 72,265 | 69,976 | 62,743 | 55,606 |
| DF4 | 72,865 | 65,298 | 63,578 | 56,522 | 50,639 |
| DF5 | 82,422 | 74,122 | 71,857 | 64,411 | 57,697 |

1C and I). Additionally, 12,170 bacterial OTUs and 6,875 fungal OTUs were identified, with 2,689 core bacterial OTUs and 1,459 core fungal OTUs shared among control and diarrheic groups.

## Microbial diversities in healthy and diarrheic foals

Alpha diversity analyses, encompassing ACE, Chao-1, Simpson, and Shannon indexes, are widely employed to gauge microbial abundance and diversity. In this study, statistical analysis revealed significant differences in bacterial ACE ($2540.45 \pm 106.58$ versus $2342.03 \pm 143.51$, $P = 0.038$), Chao-1 ($2476.76 \pm 106.95$ versus $2279.24 \pm 143.76$, $P = 0.039$), and Shannon ($8.97 \pm 0.09$ versus $8.73 \pm 0.18$, $P = 0.025$) indexes between CF and DF groups (Fig. 2A through D). Similarly, fungal Simpson ($0.99 \pm 0.01$ versus $0.97 \pm 0.01$, $P = 0.002$) and Shannon ($8.54 \pm 0.11$ versus $7.40 \pm 0.34$, $P < 0.001$) indexes showed significant decreases among DF groups (Fig. 2I through L). These results from alpha diversity analysis intuitively demonstrated that the gut microbial diversities significantly decreased during diarrhea, both in bacterial and fungal aspects. Furthermore, beta-diversity analyses, including PCoA, UPGMA, and ANOSIM, were applied to evaluate group variability. Both weighted and unweighted PCoA scatterplots illustrated that the individuals in the CF group clustered together and separated from those in the DF group, which was in line with the UPGMA tree analysis (Fig. 2E through G and M through O). ANOSIM analysis further demonstrated that the principal compositions of both bacterial ($R = 0.512$, $P = 0.010$) and fungal ($R = 0.844$, $P = 0.012$) communities were significantly distinct between the two groups (Fig. 2H and P).

## Comparison of gut bacterial and fungal communities

The gut bacterial communities were initially visualized at the phylum and genus levels. A total of 40 phyla were identified across the 10 samples, with the dominant phyla in both groups being *Firmicutes* (CF = 15.25%, DF = 19.34%), *Bacteroidota* (CF = 7.80%, DF = 13.78%), *Proteobacteria* (CF = 6.63%, DF = 5.42%), *Actinobacteriota* (CF = 2.77%, DF =

**TABLE 2** The fungal sequence data of each sample

| Sample | Raw reads | Clean reads | Denoised reads | Merged reads | Effective reads |
|--------|-----------|-------------|----------------|--------------|-----------------|
| CF1 | 51,760 | 46,953 | 46,793 | 46,224 | 45,434 |
| CF2 | 70,692 | 63,950 | 63,717 | 62,969 | 61,467 |
| CF3 | 53,070 | 47,721 | 47,507 | 46,416 | 45,711 |
| CF4 | 79,905 | 72,667 | 72,239 | 71,078 | 68,906 |
| CF5 | 80,040 | 72,129 | 71,809 | 71,097 | 69,434 |
| DF1 | 85,201 | 77,369 | 77,194 | 76,394 | 74,487 |
| DF2 | 79,872 | 72,404 | 72,238 | 71,636 | 70,289 |
| DF3 | 80,051 | 73,121 | 73,003 | 72,135 | 70,346 |
| DF4 | 79,710 | 70,476 | 70,334 | 69,581 | 68,231 |
| DF5 | 79,921 | 73,224 | 73,035 | 72,332 | 70,957 |

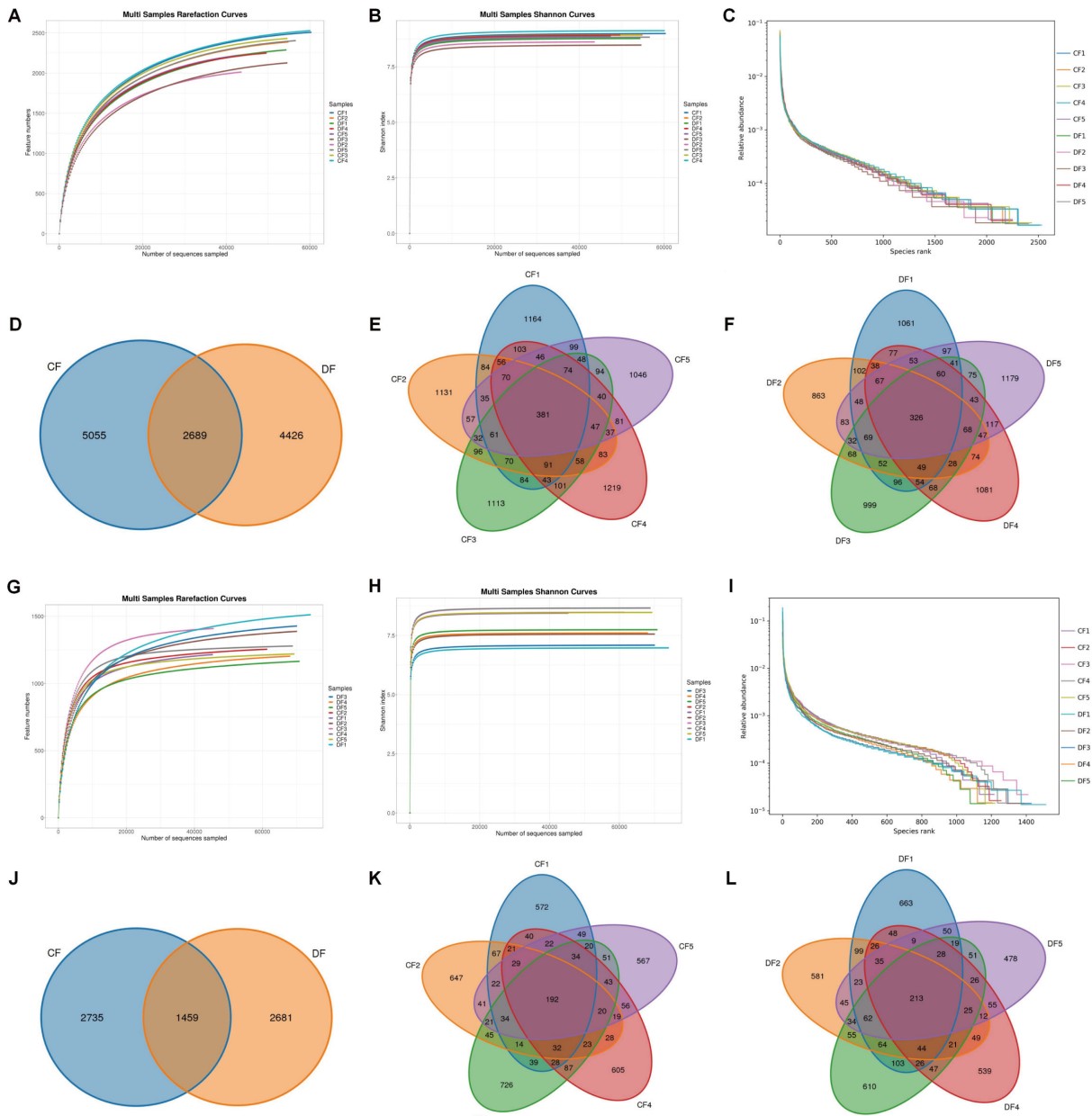

**FIG 1** Feasibility analysis of sequencing information and OTUs distribution. (A) Bacterial rarefaction curves. (B) Bacterial Shannon curves. (C) Bacterial Rank-Abundance curves. (D–F) Bacterial OTUs distribution. (G) Fungal rarefaction curves. (H) Fungal Shannon curves. (I) Fungal Rank-Abundance curves. (J–L) Fungal OTUs distribution. Both rarefaction curves and Shannon curves exhibited a tendency to reach saturation. Rank-abundance curves of all samples displayed a wide length and slow decline. Additionally, OTUs are distributed across different groups and samples.

2.41%), and *Acidobacteriota* (CF = 2.48%, DF = 2.03%), irrespective of health status. To further evaluate the changes of gut bacterial compositions during diarrhea, we identified a total of 822 bacterial genera. Among these, *unclassified_Lachnospiraceae* (2.38%), *unclassified_Archaea* (1.93%), *unclassified_Muribaculaceae* (1.83%), *unclassified_Desulfovibrionaceae* (1.71%), and *Helicobacter* (1.37%) were predominantly abundant in the CF group, whereas the dominant genera in the DF group were *unclassified_Muribaculaceae* (6.16%), *Dubosiella* (2.25%), *unclassified_Archaea* (1.90%), *unclassified_Desulfovibrionaceae* (1.87%), and *unclassified_Lachnospiraceae* (1.82%) (Fig. 3A and B).

The gut fungal compositions were also assessed, revealing 15 phyla and 729 genera across the 10 samples. The phyla *Ascomycota* (CF = 66.21%, DF = 59.78%) were most

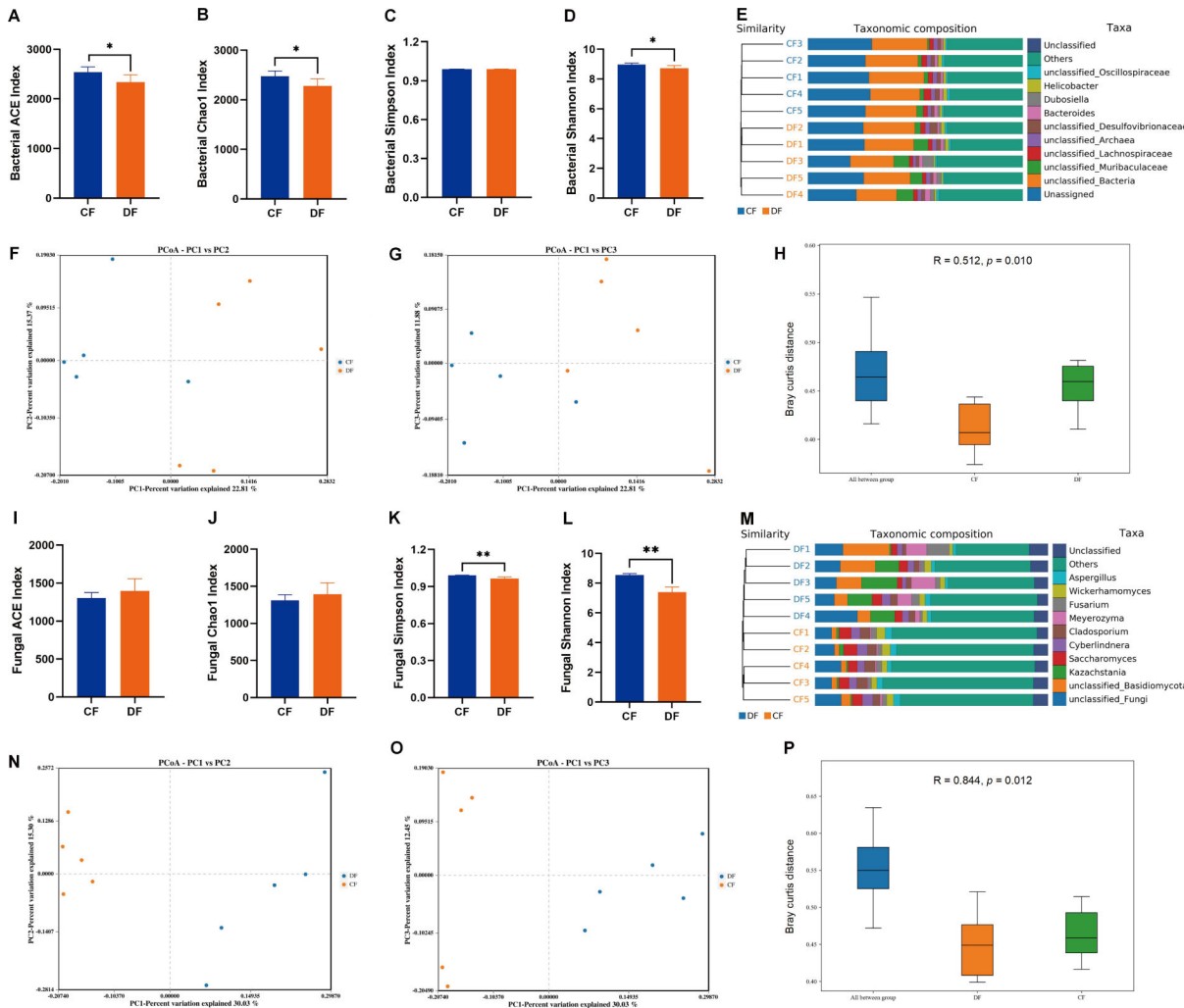

**FIG 2** Comparative analysis of the gut microbial diversities. (A) Bacterial ACE index. (B) Bacterial Chao1 index. (C) Bacterial Simpson index. (D) Bacterial Shannon index. (E) Bacterial clustering analysis. (F) Bacterial Weighted UniFrac PCoA plots. (G) Bacterial Unweighted UniFrac PCoA plots. (H) Bacterial ANOSIM analysis (I–P) Fungal diversity analysis. All of the data represent means ± SD. *$P < 0.05$, **$P < 0.01$. The bacterial ACE, Chao1, and Shannon indexes, as well as the fungal Simpson and Shannon indexes, were significantly higher in the CF group compared with the DF group. Both PCoA scatterplots and UPGMA tree analysis demonstrated that individuals in the CF group formed distinct clusters, separate from those in the DF group.

abundantly present in both groups, followed by *Basidiomycota* (CF = 15.86%, DF = 20.42%), *Chytridiomycota* (CF = 2.93%, DF = 4.95%), *Mucoromycota* (CF = 2.16%, DF = 1.61%), and *Mortierellomycota* (CF = 1.95%, DF = 0.91%). The dominant genera in the CF group were *unclassified_Basidiomycota* (11.50%), *Kazachstania* (9.36%), *Meyerozyma* (5.61%), *Fusarium* (3.68%), and *Saccharomyces* (3.15%), whereas the dominant genera in the DF group were *Saccharomyces* (4.69%), *Cladosporium* (4.00%), *Cyberlindnera* (3.89%), *Wickerhamomyces* (2.94%), and *unclassified_Basidiomycota* (2.61%) in a descending order (Fig. 3C and D).

## Significant alterations of bacteria and fungi at different taxonomic levels

Metastats analysis was conducted to characterize the specific changes in different bacterial taxa. At the phylum level, *Proteobacteria* ($P < 0.001$) and *Chloroflexi* ($P = 0.019$) were significantly more abundant in the CF group, whereas *Firmicutes* ($P = 0.046$) and *Bacteroidota* ($P = 0.013$) were notably more abundant in the DF group (Fig. 4A). Compared with the CF group, the DF group exhibited a distinct decline in the richness of 30 bacterial genera, including *Bifidobacterium* ($P = 0.018$), *Roseburia* ($P <$

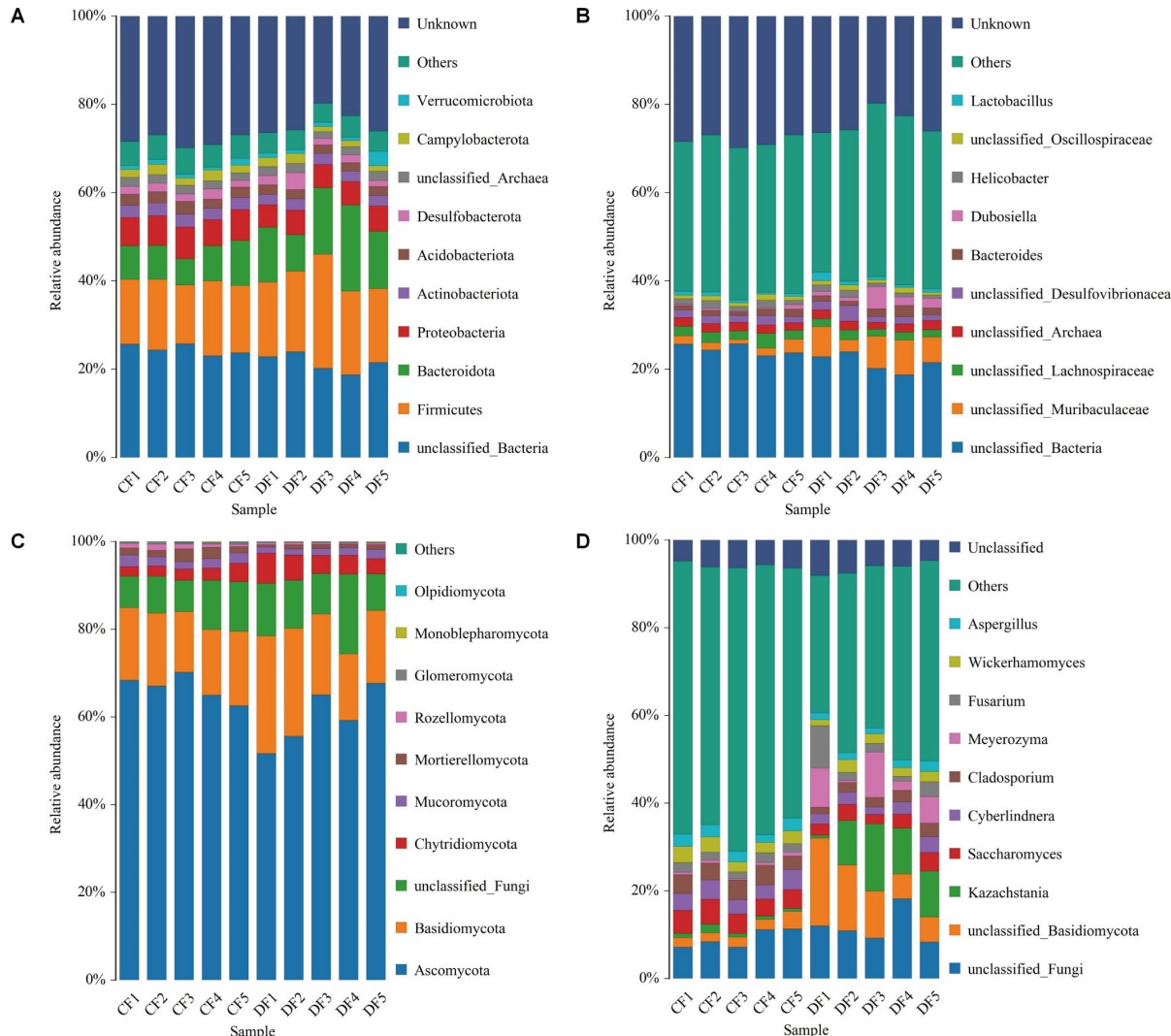

**FIG 3** The relative abundance and distributions of preponderant bacteria and fungi at different taxonomic levels. (A) Gut bacterial composition at the phylum level. (B) Gut bacterial composition at the genus level. (C) Gut fungal composition at the phylum level. (D) Gut fungal composition at the genus level.

0.001), *Prevotella_9* (*P* = 0.046), *Coprococcus* (*P* = 0.005), *Lachnospiraceae_FCS020_group* (*P* = 0.043), and *Terrisporobacter* (*P* = 0.037), along with a significant increase in the abundance of six genera, involving *Monoglobus* (*P* = 0.038), *Adlercreutzia* (*P* = 0.033), *Candidatus_Saccharimonas* (*P* = 0.008), *Enterorhabdus* (*P* < 0.001), *Romboutsia* (*P* = 0.010), and *Dubosiella* (*P* = 0.045) (Fig. 4A and B). Additionally, LEfSe analysis was employed to ensure comprehensive identification of potential bacterial biomarkers; however, no additional biomarkers were identified (Fig. 6A and B).

The significant differences in fungal taxa were also investigated using Metastats analysis. At the phylum level, we observed an increase in the abundance of *Chytridiomycota* (*P* = 0.012) and a decrease in *Ascomycota* (*P* = 0.042) during diarrhea (Fig. 5A). Compared with the control foals, diarrheic foals exhibited a decline in the richness of 34 genera, including *Saccharomyces* (*P* = 0.006), *Cyberlindnera* (*P* = 0.010), *Pichia* (*P* = 0.002), and *Ophiocordyceps* (*P* = 0.041), as well as an increase in the abundance of four genera, involving *Acrocalymma* (*P* = 0.025), *Fodinomyces* (*P* = 0.045), *Nigrospora* (*P* = 0.015), and *unclassified_Dipodascaceae* (*P* = 0.040) (Fig. 5A and B). Additionally, LEfSe analysis was applied to investigate fungal biomarkers associated with diarrhea. Besides the aforementioned fungal taxa, *Basididmycota* was also found to be more dominant in the DF group (Fig. 6C and D).

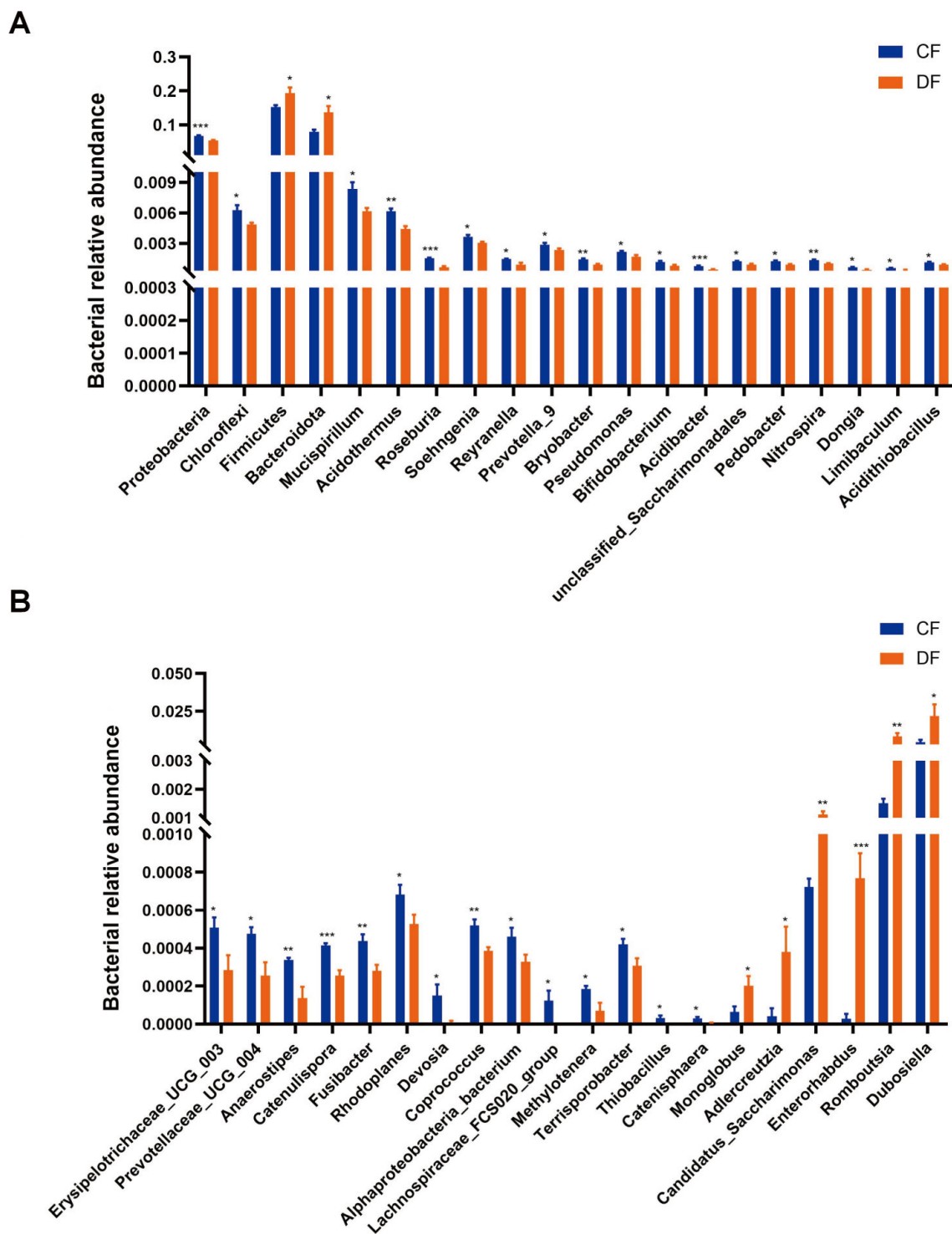

**FIG 4** Significant alterations of bacteria at phylum and genus levels. (A) Significant alterations of four bacterial phyla and 16 genera. (B) Significant alterations of the remaining 20 bacterial genera. $*P < 0.05$, $**P < 0.01$, $***P < 0.001$.

## Correlation network analysis

A network analysis was performed to reveal the correlations between different bacterial genera. We observed that *Dubosiella*, one of the biomarkers of the DF group, exhibited positive correlations with *Bacteroides* (0.7818), *Faecalibaculum* (0.7697), and *Parabacteroides* (0.7697). *Acidothermus,* one of the biomarkers of CF group, displayed

positive correlations with *Mucispirillum* (0.8424), *Soehngenia* (0.7818), and *Paraclostridium* (0.7818). Additionally, *Dubosiella* and *Acidothermus* showed a negative correlation (−0.7333) (Fig. 7A).

*Saccharomyces*, a fungal genus preponderant in the CF group, was positively correlated with *Aspergillus* (0.9152), *Cyberlindnera* (0.9030), *Naumovozyma* (0.8424), *Wickerhamomyces* (0.8303), and *Trichosporon* (0.8303). *Pichia*, another biomarker of

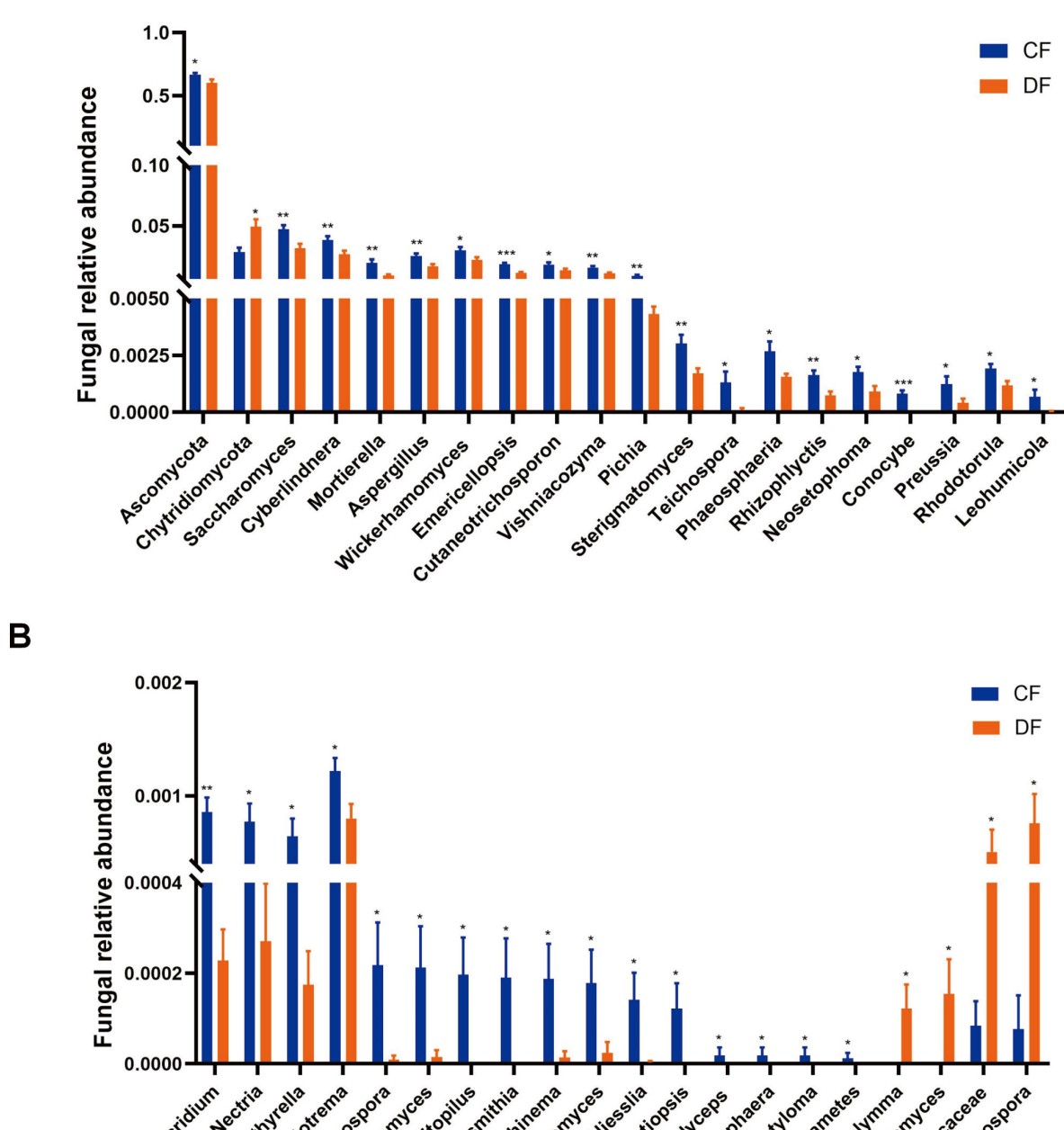

**FIG 5** Significant alterations of fungi at phylum and genus levels. (A) Significant alterations of two fungal phyla and 18 genera. (B) Significant alterations of the remaining 20 fungal genera. *P < 0.05, **P < 0.01, ***P < 0.001.

CF group, was positively correlated with *Sampaiozyma* (0.9152), *Clavispora* (0.8788), *Alternaria* (0.8303), *Plectosphaerella* (0.8182), and *Filobasidium* (0.8182) (Fig. 7B).

## DISCUSSION

Diarrhea is one of the most widespread ailments affecting farm animals, posing a significant challenge to animal welfare and husbandry practices. Foals, in particular, are highly susceptible to diarrhea, with up to 60% experiencing the condition within the first 6 months of life, significantly contributing to their high morbidity and mortality rates (2, 3, 32). Managing diarrhea in foals is challenging due to factors such as stress responses and antibiotic resistance. Nonetheless, delving into the intricacies of the gut microbiota and identifying pertinent biomarkers hold promise for developing preventive and therapeutic interventions. Comprehensive studies have shown that maintaining a stable gut microbiota is essential for the immune system, metabolism, and intestinal barrier, and an imbalance in the gut microbiota can lead to several diseases (1, 33–37). Previous studies have highlighted the association between diarrhea and gut microbiota across various species, including piglets, giraffes, yaks, and goats (9, 10, 30, 38). However, research on gut microbial alterations in diarrheic foals, particularly focusing on fungal aspects, is limited.

The gut microbiota undergoes dynamic changes to adapt to the physiological variations of its hosts, and these alterations are typically limited in scope (39). However, numerous factors can significantly disrupt this natural balance, including antibiotics, microplastics, and several diseases (diarrhea being among the most extensively researched) (1, 40–42). For instance, He et al. (43) reported a significant alteration in

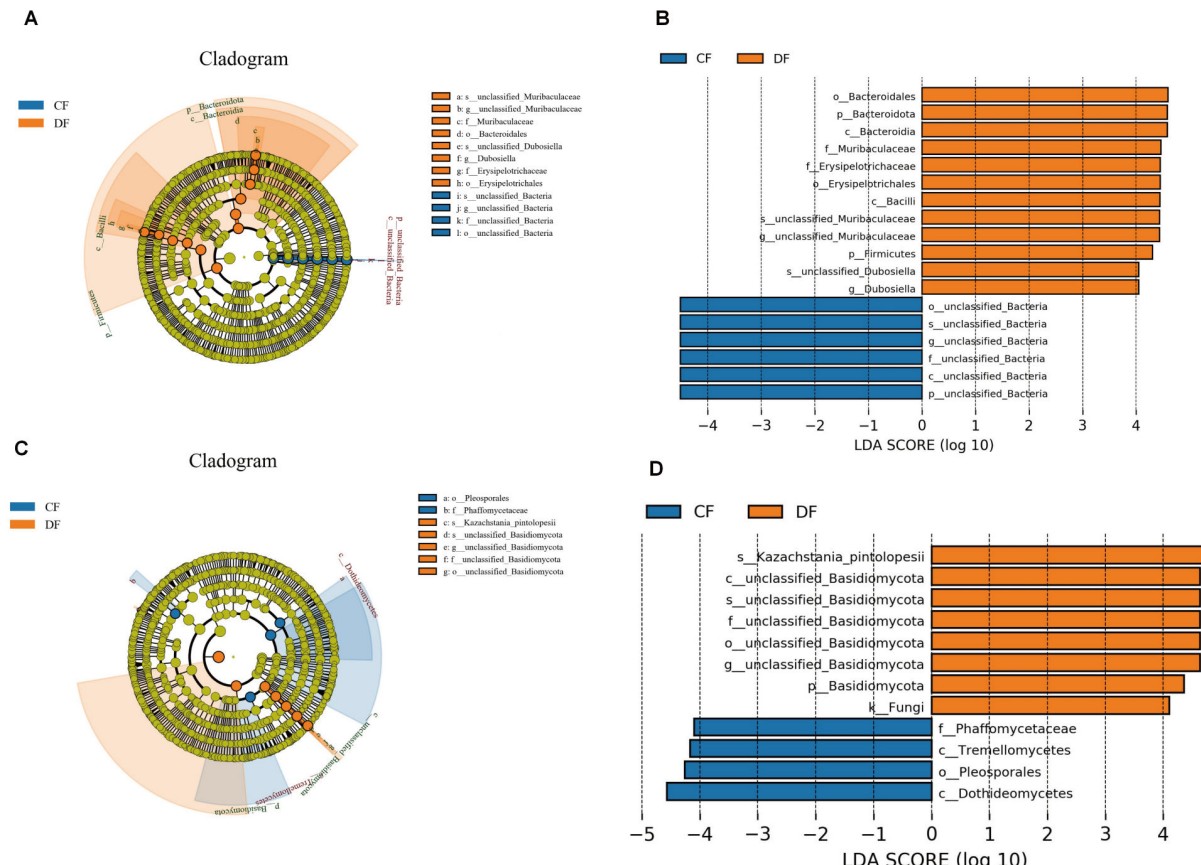

**FIG 6** Diagram illustrating the variance in the abundance of bacteria and fungi based on LEfSe analysis and LDA scores. Cladogram depicting the phylogenetic distribution of intestinal bacteria (A) and fungi (C) associated with each group. Significant differences in the relative abundance of bacteria (B) and fungi (D) between the healthy and the diarrheic groups. LDA scores greater than four were considered statistically significant.

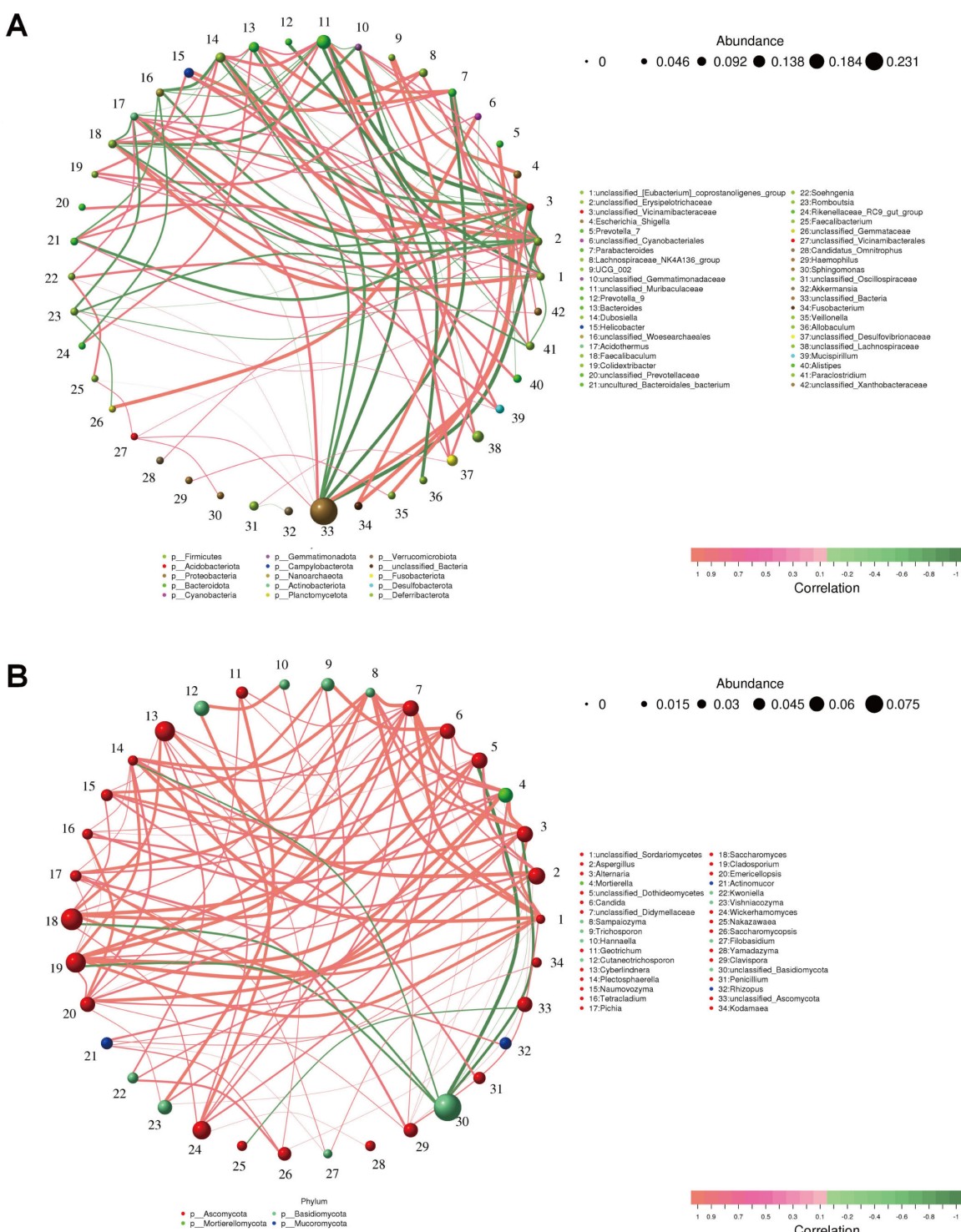

**FIG 7** Correlation network analysis. (A) Correlations among different bacterial genera. (B) Correlations among different fungal genera. The circles of varying colors represent distinct bacterial and fungal genera, with their respective sizes reflecting the relative abundance. The thickness of the connecting lines indicates the strength of the correlation between these genera. Green lines denote positive correlations, whereas pink lines signify negative correlations.

the diversity of piglet fecal microbiota during diarrhea. Liu et al. demonstrated that the diarrheic yaks presented decreased bacterial and fungal diversity compared to healthy populations (10). Consistent with these findings, our research revealed a substantial decline in bacterial and fungal diversity among diarrheic foals. Moreover, the principal

coordinate analysis indicated a distinct difference in the primary composition of gut bacterial and fungal communities between the two groups. It highlighted that despite sharing the same environment and diets, foals exhibited notable alterations in their gut microbiota structure during diarrhea. The intestine is the primary site for absorption, which relies on normal intestinal morphology and a healthy gut microbial composition (43, 44). An imbalance in gut microbial composition can play a pivotal role in the development of various diseases (45–47). Furthermore, such dysbiosis could lead to increased intestinal permeability and weakened immunity, thereby facilitating the infiltration of pathogenic microbes and conditioned pathogens into the system (44, 48, 49). Previous studies have indicated that gut microbial dysbiosis may contribute to the high mortality rates observed in diarrheic goats and yaks (10, 38). Similarly, the gut microbial imbalance observed in our research may also be responsible for the high mortality rate of diarrheic foals.

In this study, *Firmicutes* and *Bacteroidota* were identified as the most dominant bacterial phyla, whereas *Ascomycota* and *Basidiomycota* were noted as the most prevalent fungal phyla in foals, regardless of the presence of diarrhea. Interestingly, these microbial phyla were also prominently present in the gut microbiota of other herbivores such as cattle, goats, giraffes, and yaks, underscoring their significance in shaping intestinal ecology and functions within herbivorous animals (5, 8–10). Although the dominant phyla remained consistent, their relative abundances exhibited significant changes. In foals with diarrhea, we observed a notable reduction in the proportions of two bacterial phyla (*Proteobacteria* and *Chloroflexi*) and one fungal phylum (*Ascomycota*), alongside an increase in the proportions of two bacterial phyla (*Firmicutes* and *Bacteroidota*) and one fungal phylum (*Chytridiomycota*). *Proteobacteria*, a phylum encompassing diverse gram-negative bacteria, have been closely linked to energy accumulation in humans, monkeys, and mice (50–53). Although constituting a relatively minor portion of the gut microbiota, *Proteobacteria* play various roles in metabolism, fermentation, and immune modulation, all of which are essential for maintaining gut homeostasis and host health (54). Consistent with our findings, a similar decrease in the relative abundance of *Proteobacteria* was observed in diarrheic yaks and adult horses (10, 55). *Ascomycota*, which account for over half of the gut fungi in foals, are important in dietary fiber degradation and immune modulation (56, 57). Given that foals are strict herbivores requiring substantial amounts of forage to fulfill their energy and growth needs, the decrease in the abundance of these bacterial and fungal phyla may be a crucial factor contributing to weight loss and high mortality observed in foals with diarrhea.

An in-depth analysis was conducted to examine the microbial variations at the genus level, as identifying specific microorganisms could offer insights into potential correlations between the gut microbiota and the onset of diarrhea. We identified significant variations in 36 bacterial genera, with 30 experiencing significant downregulation in response to diarrhea. Notably, two altered genera were undetectable in the samples from diarrheic foals. This suggests that the intestinal conditions in diarrheic foals may undergo notable changes, leading to selective inhibition of certain bacteria colonization. Among the 30 decreased bacterial genera, *Bifidobacterium, Mucispirillum, Coprococcus, Prevotella_9, Catenisphaera, Roseburia, Terrisporobacter,* and *Anaerostipes* are recognized as beneficial intestinal bacteria that play crucial roles in digestion, metabolism, immunity, and overall gut health. *Bifidobacterium*, a natural colon inhabitant, is widely recognized as a probiotic. Studies have demonstrated its potential to enhance glucose and lipid metabolism, bolstering neonatal immunity and improving cognitive flexibility (58–60). Additionally, *Bifidobacterium* has been reported to effectively alleviate fat deposition, infant colic, and constipation (61–63). *Mucispirillum* plays a vital role in gut health and mucosal homeostasis and has been reported to be effective in antagonizing *Salmonella* virulence, thereby protecting against colitis in mice (64). *Coprococcus* can alleviate colitis by mediating the IgA response and restoring the gut microbiota (65). The genus *Roseburia*, comprising five gram-positive obligate anaerobic bacteria, produces butyrate in the colon, influencing colonic motility and immunity

(66). Numerous research reported that *Roseburia* can reduce the risk of intestinal-related diseases, including Crohn's disease, colorectal cancer, gut-dysbiosis-induced mastitis, and ulcerative colitis (67–70). A decreased abundance of Prevotella_9 has been linked to depression and anxiety in patients with active ulcerative colitis (71). However, a higher abundance of *Catenisphaera* in the gut bacterial community has been found to be beneficial in alleviating weaning stress in lambs (72). Recent investigations have shed light on the pivotal function of short-chain fatty acids (SCFAs) in mechanisms against cancer and inflammation and in regulating energy metabolism (73, 74). *Terrisporobacter* and *Anaerostipes* are potential producers of SCFAs, playing a dual role in enhancing host immunity and preventing the colonization of pathogenic bacteria by modulating intestinal pH (75, 76). Notably, none of the 36 significantly altered bacterial genera included typical pathogenic bacteria.

Fungi constitute a crucial component of the gut microbiota, significantly contributing to the intestinal ecosystem and host health, particularly in herbivores (77, 78). Similar to the changes observed in gut bacteria, the taxonomic composition of the gut fungal community underwent significant alterations during diarrhea, characterized by an increase in the abundance of four fungal genera and a decline of 34 fungal genera. Furthermore, eight decreased fungi were even undetectable in the samples from diarrheic foals, implying their inability to adapt to the current intestinal environment. Among decreased fungi, *Ophiocordyceps, Saccharomyces, Rhodotorula, Mortierella, Aspergillus,* and *Cyberlindnera* are considered beneficial due to their roles in fermentation, nutrient absorption, and host health. *Ophiocordyceps* is a genus of fungi known for its unique life cycle and potential medicinal properties. One of the most well-known members of this genus is *Ophiocordyceps sinensis*, commonly referred to as the caterpillar fungus, which is highly prized in traditional Chinese medicine and has been used for centuries. Previous studies indicated that *Ophiocordyceps* species and their extracts can attenuate multiple diseases, including chronic kidney disease, pulmonary fibrosis, and Hashimoto's thyroiditis (79–81). *Saccharomyces*, a genus including various yeast species, many of which are important in fermentation, bioengineering, and healthcare, features *Saccharomyces boulardii*, a successful probiotic yeast often used for diarrhea treatment (82). As a potential gut microbiota modulator, *Saccharomyces boulardii* can control gastrointestinal tract disorders by alleviating inflammation and improving intestinal barrier dysfunction (83, 84). Certain strains of *Rhodotorula* exhibit robust lipid and carotene production capabilities (85). Although some *Aspergillus* species can cause health issues in animals, others are known for their role in antibiotic production. *Cyberlindnera* is recognized for its capacity to produce diverse valuable compounds used in the food and pharmaceutical industries (86). Recent research has demonstrated that dietary supplementation with *Cyberlindnera jadinii* can enhance the growth performance and intestinal health of various farm animals, including dairy cows, broiler chickens, and weaned piglets (87–89). Although gut bacterial communities have long been implicated as contributors to diarrhea, only a few studies have highlighted fungal communities' involvement in diarrhea among giraffes and yaks (9, 10). In this study, we investigated the shifts in the gut fungal community in diarrheic foals, emphasizing the significance of fungal community dysbiosis in foal diarrhea.

The stability of gut microbiota is essential for preventing diseases, inhibiting pathogens, and maintaining intestinal barrier integrity, although its disruption is considered a contributing factor in the pathology of numerous diseases (45–47, 90). Gut bacteria and fungi engage in various commensal, symbiotic, or antagonistic interactions to establish a stable ecosystem (22). Consequently, changes in certain intestinal bacteria and fungi can impact other species, leading to significant alterations in the gut microbiota. The correlation network analysis of this study unveiled notable connections between the modified bacteria and fungi. Importantly, these altered microorganisms were observed to influence certain bacteria and fungi that did not undergo significant changes during diarrhea through microbial interactions, thereby amplifying the influence of diarrhea on the gut microbiota and intestinal functions. These findings

suggest that diarrhea directly altered the diversities and compositions of gut bacterial and fungal communities and indirectly affected some microorganisms through microbial interactions, potentially leading to further disruption of intestinal homeostasis. Considering the significant reduction of beneficial microbes observed in this study, targeted probiotic supplementation could serve as a preventative and therapeutic approach for managing foal diarrhea. Several studies have previously reported that supplementation with *Bacillus cereus* and *Lactobacillus rhamnosus* does not improve foal diarrhea (91, 92). However, supplementing with beneficial genera identified in this study as significantly reduced during foal diarrhea, such as *Bifidobacterium* and *Saccharomyces*, may yield different outcomes. Future research should focus on isolating and testing specific probiotic strains identified in this study, assessing their efficacy in promoting gut microbial stability, enhancing host immunity, and reducing the incidence and severity of diarrhea in foals.

## Conclusion

The study elucidates the alterations in gut microbial composition in foals during diarrhea. The results revealed a significant decrease in both bacterial and fungal diversities in diarrheic foals, alongside notable alterations in gut microbial composition. This research advances our understanding of gut bacterial and fungal dynamics in foals under varying health conditions, emphasizing gut microbiota dysbiosis as a potential contributor to foal diarrhea. Moreover, we observed that many altered bacteria and fungi, including some probiotic strains, were downregulated during diarrhea. This suggests that probiotics may offer superior efficacy compared with antibiotics as potential candidates for preventing and treating foal diarrhea. However, further studies with larger sample sizes conducted across diverse regions are necessary to enhance the generalizability and applicability of these findings. Additionally, the probiotics identified in this study should be further investigated to support their application in equine health management.

## ACKNOWLEDGMENTS

This study acknowledges the Knowledge Innovation Program of Wuhan-Shuguang Project (2023020201020351).

D.Z.: Formal analysis, Writing – original draft. S.L.: Investigation, Validation. Z.X.: Data curation. M.F.K.: Writing – review & editing. X.B.: Resources. Y.W.: Resources. B.W.: Visualization. E.K.: Writing – review & editing. D.D.: Software. L.W.: Software. Y.C.: Software. A.G.: Writing – review & editing. Y.S.: Funding acquisition, Project administration, Supervision.

## AUTHOR AFFILIATIONS

[1]College of Veterinary Medicine, Huazhong Agricultural University, Wuhan, China
[2]China Horse Industry Association, Beijing, China

## AUTHOR ORCIDs

Yaoqin Shen http://orcid.org/0000-0002-5774-7688

## FUNDING

| Funder | Grant(s) | Author(s) |
| --- | --- | --- |
| Knowledge Innovation Program of Wuhan-Shuguang Project | 2023020201020351 | Yaoqin Shen |

## AUTHOR CONTRIBUTIONS

Di Zhu, Formal analysis, Writing – original draft | Siyu Li, Investigation, Validation | Zhixiang Xu, Data curation | Md. F. Kulyar, Writing – review and editing | Xu Bai, Resources | Yu Wang, Resources | Boya Wang, Visualization | Emaan Khateeb, Writing – review and editing | Dandan Deng, Software | Lidan Wang, Software | Yuji Chen, Software | Aizhen Guo, Writing – review and editing | Yaoqin Shen, Funding acquisition, Project administration, Supervision

## DATA AVAILABILITY

Amplicon sequences are available under BioProject. The raw FASTQ files have been deposited in the NCBI Sequence Read Archive (SRA) with the accession ID number PRJNA1082084.

## ETHICS APPROVAL

The animal study was performed under the guidance of the Ethical Committee of the Huazhong Agricultural University, which was in accordance with the local legislation and institutional requirements.

## ADDITIONAL FILES

The following material is available online.

Open Peer Review

PEER REVIEW HISTORY (review-history.pdf). An accounting of the reviewer comments and feedback.

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
