## [Reviewer comments · Microbiology Spectrum]

Microbiology Spectrum

Comparative Analysis of Gut Microbiota in Healthy and Diarrheic Foals

Di Zhu, Siyu Li, Zhixiang Xu, Muhammad Fakhar-e-Alam Kulyar, Xu Bai, Yu Wang, Boya Wang, Emaan Khateeb, Dandan Deng, Lidan Wang, Yuji Chen, Aizhen Guo, and Yaoqin Shen

Corresponding Author(s): Yaoqin Shen, Huazhong Agricultural University

Review Timeline:

Submission Date:	April 5, 2024
Editorial Decision:	August 4, 2024
Revision Received:	August 19, 2024
Editorial Decision:	November 24, 2024
Revision Received:	January 18, 2025
Accepted:	February 1, 2025

Editor: Francisco Uzal

Reviewer(s): Disclosure of reviewer identity is with reference to reviewer comments included in decision letter(s). The following individuals involved in review of your submission have agreed to reveal their identity: Yuchao Zhao (Reviewer #2)

Transaction Report:

DOI: <https://doi.org/10.1128/spectrum.00871-24>

Re: Spectrum00871-24 (Comparative analysis of gut microbiota in healthy and diarrheic foals)

Dear Dr. Yaoqin Shen:

Thank you for the privilege of reviewing your work. Below you will find my comments, instructions from the Spectrum editorial office, and the reviewer comments.

Revision Guidelines

Sincerely,
Francisco Uzal
Editor
Microbiology Spectrum

Reviewer #1 (Comments for the Author):

The manuscript entitled "Comparative Analysis of Gut Microbiota in Healthy and Diarrheic Foals" explored the changes in gut microbiota in foals during diarrhea. This study holds significant implications for understanding the gut microbiota of foals, particularly given the limited research on foal diarrhea. However, several issues need to be addressed before publication. Below are some specific comments:

1. Please revise the language throughout the manuscript for clarity and precision. For example, change "with" to "during" in line

- 12, and "microbes" to "gastrointestinal microorganisms" in line 48.
2. Lines 123 through 137 should be formatted using two-end alignment for better readability.
3. Provide more detail on how samples were collected to minimize contamination. This is crucial as contamination can significantly impact microbiota studies.
4. The manuscript lacks specificity regarding the handling of diarrheic foals prior to sample collection. Given that treatment was withheld until after sampling, it is critical to disclose whether this delay could have influenced the microbiota composition, potentially confounding the results.
5. Using the TGuide S96 Magnetic Stool DNA Kit for DNA extraction appears appropriate, with standard quality control measures in place. However, the manuscript lacks information on the specific conditions under which DNA extraction and sequencing were performed, which is crucial for assessing potential biases in the data.
6. To ensure transparency and reproducibility, further clarification is needed regarding the parameters and thresholds applied during data processing, such as the exact cutoffs for OTU clustering and statistical significance.
7. Provide more detailed explanations for the figures because clear, well-labeled diagrams are essential for effectively conveying the study's findings.
8. Provide p-values for the section on "Significant Alterations of Bacteria and Fungi at Different Taxonomic Levels" to support the statistical significance of the findings.
9. Discuss the study's limitations in the "Discussion" or "Conclusion" section. Acknowledging the limitations will provide a more balanced perspective on the findings.
10. Address whether five samples per group are sufficient for omics analysis.

Dear Reviewer,

Microbiology Spectrum

We are grateful for your thorough review of our manuscript entitled "Comparative Analysis of Gut Microbiota in Healthy and Diarrheic Foals" (Spectrum00871-24). Your comments have been invaluable in refining our work. We have addressed each point you raised as detailed below (highlighted in blue color).

Reviewer #1: The manuscript entitled "Comparative Analysis of Gut Microbiota in Healthy and Diarrheic Foals" explored the changes in gut microbiota in foals during diarrhea. This study holds significant implications for understanding the gut microbiota of foals, particularly given the limited research on foal diarrhea. However, several issues need to be addressed before publication. Below are some specific comments:

1. Please revise the language throughout the manuscript for clarity and precision. For example, change "with" to "during" in line 12, and "microbes" to "gastrointestinal microorganisms" in line 48.

Response: Thank you for your constructive suggestion. We have revised the language throughout the manuscript, especially in the sections you mentioned.

2. Lines 123 through 137 should be formatted using two-end alignment for better readability.

Response: Thank you for your kind reminder. We have formatted this paragraph using two-end alignment (lines 154-167).

3. Provide more detail on how samples were collected to minimize contamination. This is crucial as contamination can significantly impact microbiota studies.

Response: We appreciate this suggestion. We have added more detailed information on how samples were collected to minimize contamination (line 94-97).

4. The manuscript lacks specificity regarding the handling of diarrheic foals prior to sample collection. Given that treatment was withheld until after sampling, it is

critical to disclose whether this delay could have influenced the microbiota composition, potentially confounding the results.

Response: Respected reviewer, we agree that delayed treatment can influence the microbiota composition of diarrheic foals. To address this concern, we selected Yumayuan Scenic Area as our sampling site due to the availability of a large number of foals meeting our sampling standards. In fact, only one day elapsed between diagnosis and sampling. Given the prolonged duration of foal diarrhea, we believe that the samples we collected are representative to a large extent.

5. Using the TGuide S96 Magnetic Stool DNA Kit for DNA extraction appears appropriate, with standard quality control measures in place. However, the manuscript lacks information on the specific conditions under which DNA extraction and sequencing were performed, which is crucial for assessing potential biases in the data.

Response: Thank you for your constructive suggestion. We have provided more detailed information on the specific conditions under which DNA extraction (line 103-106) and sequencing (line 115-121) were performed.

6. To ensure transparency and reproducibility, further clarification is needed regarding the parameters and thresholds applied during data processing, such as the exact cutoffs for OTU clustering and statistical significance.

Response: Thank you for your kind reminder. We have provided more detailed information on the thresholds applied during data processing and statistical analysis (line 145-152).

7. Provide more detailed explanations for the figures because clear, well-labeled diagrams are essential for effectively conveying the study's findings.

Response: We appreciate your valuable feedback. We have enhanced the clarity of the figures by providing more detailed explanations (line 418-458).

8. Provide p-values for the section on "Significant Alterations of Bacteria and Fungi at Different Taxonomic Levels" to support the statistical significance of the findings.

Response: Thank you for your kind reminder. We have added all *p*-values to

support the statistical significance of the findings in this section (line 210-235).

9. Discuss the study's limitations in the "Discussion" or "Conclusion" section. Acknowledging the limitations will provide a more balanced perspective on the findings.

Response: Thank you for your constructive suggestion. We have revised the conclusion section to include the limitations of the study, and suggested directions for future research in this section (line 412-416).

10. Address whether five samples per group are sufficient for omics analysis.

Response: Respected reviewer, the consistency and significance of our results indicate that five samples per group were sufficient to capture the key biological differences and trends. However, we acknowledge that a larger sample size could further strengthen the statistical power of our findings. As such, we have outlined directions for future research that will include larger sample sizes and additional regions for analysis in the conclusion section.

Finally, we would like to extend our sincere appreciation for such nice and resourceful feedback. Indeed, your insights have been invaluable in improving the quality of our manuscript. We are truly grateful for your meticulous perusal of our manuscript and for taking valuable time to appraise our work.

Kind regards,

Assoc. Prof. Yaoqin Shen

College of Veterinary Medicine,

Huazhong Agricultural University, Wuhan 430070, PR China

Telephone: +86-152-7189-8678

Email: yshen@mail.hzau.edu.cn

Re: Spectrum00871-24R1 (Comparative Analysis of Gut Microbiota in Healthy and Diarrheic Foals)

Dear Dr. Yaoqin Shen:

Thank you for the privilege of reviewing your work. Below you will find my comments, instructions from the Spectrum editorial office, and the reviewer comments.

Revision Guidelines

Sincerely,
Francisco Uzal
Editor
Microbiology Spectrum

Reviewer #1 (Comments for the Author):

I believe that paper is in better shape than the previous version of the paper

Reviewer #2 (Comments for the Author):

The authors compared the fecal microbiota differences between healthy foals and those with diarrhea. The manuscript is well-organized, and the authors have revised it according to the reviewers' comments. However, some issues still require further improvement:

In line 42, the authors mention "preliminary studies" but only cite one reference. Please include additional citations to support this statement.

The Introduction should begin with an overview of diarrhea in foals, including the incidence rate of diarrhea.

In lines 81-82, it is crucial to address the cause of diarrhea in foals. Are the same factors responsible for diarrhea in all cases? If different factors cause diarrhea, this would present a more complex issue.

In lines 133-134, clarify which data were analyzed using R and which were analyzed using GraphPad Prism.

In Figures 3 and 4, please verify the units on the y-axis. Relative abundance should be in percentages, yet the values appear low.

In line 155, there is a typographical error; it should read "OTU" instead of "OUT." Please check the entire manuscript for similar errors, as these are common in Word.

In line 157, while performing PCoA, did the authors conduct any similarity tests, such as ANOSIM?

In lines 257-259, please correct the citation format, positioning it after the first author's name and "et al." rather than at the end of the sentence.

Is there any application of probiotics for foals? Please discuss this aspect in the Discussion section.

Dear Reviewers,

Microbiology Spectrum

We sincerely appreciate your thorough review of our manuscript, titled “Comparative Analysis of Gut Microbiota in Healthy and Diarrheic Foals” (Spectrum00871-24R1). Your constructive comments and suggestions have been invaluable in refining and improving our work. We have carefully addressed each point you raised, and our detailed responses are provided below (highlighted in blue).

Reviewer #1: I believe that paper is in better shape than the previous version of the paper.

Response: Thank you for your positive feedback. We are pleased that the revisions have improved the manuscript and addressed your concerns. Your insights have been invaluable in refining the quality of our work.

Reviewer #2: The authors compared the fecal microbiota differences between healthy foals and those with diarrhea. The manuscript is well-organized, and the authors have revised it according to the reviewers' comments. However, some issues still require further improvement:

1. In line 42, the authors mention "preliminary studies" but only cite one reference. Please include additional citations to support this statement.

Response: Thank you for your kind reminder. To strengthen our statement, we have included two additional references that further support the positive effect of FMT on diarrhea symptoms (12, 13).

2. The Introduction should begin with an overview of diarrhea in foals, including the incidence rate of diarrhea.

Response: Thank you for your suggestion. We have incorporated an overview of diarrhea in foals, including its incidence rate, at the beginning of the Introduction section (lines 40-41).

3. In lines 81-82, it is crucial to address the cause of diarrhea in foals. Are the same

factors responsible for diarrhea in all cases? If different factors cause diarrhea, this would present a more complex issue.

Response: Thank you for highlighting this point. We agree that the causes of diarrhea in foals can vary and may involve multiple factors. In response, we have clarified that all the foals in this study shared the same environment, and the diarrheic foals were diagnosed with non-infectious diarrhea (line 84-87). This type of diarrhea is common but remains under-researched in foals. Thus, our study aims to explore this condition through the lens of gut microbiota.

4. In lines 133-134, clarify which data were analyzed using R and which were analyzed using GraphPad Prism.

Response: Thank you for your kind reminder. We have addressed this point by adding, “Statistical analysis of alpha diversity and differential microorganisms was performed using GraphPad Prism (v8.0), while the remaining data were analyzed using the R package (v3.0.3).” in lines 134-136.

5. In Figures 3 and 4, please verify the units on the y-axis. Relative abundance should be in percentages, yet the values appear low.

Response: Thank you for your constructive suggestion. Upon reviewing the raw data, we discovered that the units on the y-axis in these two figures were incorrect. The “%” symbols have now been removed, and the corrected figures have been re-uploaded.

6. In line 155, there is a typographical error; it should read "OTU" instead of "OUT." Please check the entire manuscript for similar errors, as these are common in Word.

Response: Thank you for your helpful reminder. We have corrected the typographical error in line 156 and have thoroughly reviewed the entire manuscript to address similar errors to the best of our ability.

7. In line 157, while performing PCoA, did the authors conduct any similarity tests, such as ANOSIM?

Response: Thank you for your valuable feedback. We have incorporated the results of the ANOSIM analysis into Figure 2 and included a corresponding

description in line 173-175. The ANOSIM analysis demonstrated that the principal compositions of both bacterial ($R = 0.512$, $p = 0.010$) and fungal ($R = 0.844$, $p = 0.012$) communities were significantly distinct between two groups.

8. In lines 257-259, please correct the citation format, positioning it after the first author's name and "et al." rather than at the end of the sentence.

Response: Thank you for your helpful reminder. We have corrected the citation format as suggested.

9. Is there any application of probiotics for foals? Please discuss this aspect in the Discussion section.

Response: Thank you for your valuable suggestion. We have included information on the applications of probiotics for managing foal diarrhea and have discussed this aspect in detail in the Discussion section (line 378-387).

Once again, we extend our heartfelt gratitude for your insightful and resourceful feedback. Your meticulous review and thoughtful appraisal have significantly enhanced the quality of our manuscript.

Kind regards,

Assoc. Prof. Yaoqin Shen

College of Veterinary Medicine,

Huazhong Agricultural University, Wuhan 430070, PR China

Telephone: +86-15271898678

Email: yshen@mail.hzau.edu.cn

Re: Spectrum00871-24R2 (Comparative Analysis of Gut Microbiota in Healthy and Diarrheic Foals)

Dear Dr. Yaoqin Shen:

Your manuscript has been accepted, and I am forwarding it to the ASM production staff for publication. Your paper will first be checked to make sure all elements meet the technical requirements. ASM staff will contact you if anything needs to be revised before copyediting and production can begin. Otherwise, you will be notified when your proofs are ready to be viewed.

Sincerely,
Francisco Uzal
Editor
Microbiology Spectrum